# Real-Time 3D Reconstruction of Thin Surface Based on Laser Line Scanner

**DOI:** 10.3390/s20020534

**Published:** 2020-01-18

**Authors:** Yuan He, Shunyi Zheng, Fengbo Zhu, Xia Huang

**Affiliations:** School of Remote Sensing and Information Engineering, Wuhan University, Wuhan 430079, China

**Keywords:** real-time 3D reconstruction, laser line scanner, thin surface, parallel computing, GPU

## Abstract

The truncated signed distance field (TSDF) has been applied as a fast, accurate, and flexible geometric fusion method in 3D reconstruction of industrial products based on a hand-held laser line scanner. However, this method has some problems for the surface reconstruction of thin products. The surface mesh will collapse to the interior of the model, resulting in some topological errors, such as overlap, intersections, or gaps. Meanwhile, the existing TSDF method ensures real-time performance through significant graphics processing unit (GPU) memory usage, which limits the scale of reconstruction scene. In this work, we propose three improvements to the existing TSDF methods, including: (i) a thin surface attribution judgment method in real-time processing that solves the problem of interference between the opposite sides of the thin surface; we distinguish measurements originating from different parts of a thin surface by the angle between the surface normal and the observation line of sight; (ii) a post-processing method to automatically detect and repair the topological errors in some areas where misjudgment of thin-surface attribution may occur; (iii) a framework that integrates the central processing unit (CPU) and GPU resources to implement our 3D reconstruction approach, which ensures real-time performance and reduces GPU memory usage. The proposed results show that this method can provide more accurate 3D reconstruction of a thin surface, which is similar to the state-of-the-art laser line scanners with 0.02 mm accuracy. In terms of performance, the algorithm can guarantee a frame rate of more than 60 frames per second (FPS) with the GPU memory footprint under 500 MB. In total, the proposed method can achieve a real-time and high-precision 3D reconstruction of a thin surface.

## 1. Introduction

Recent years have witnessed the release of a large number of fast, accurate depth sensors. With the opening up of graphics processors to general computing, 3D reconstruction by the depth sensors has attracted new attention in the field of computer vision [1]. It has a wide range of applications in reverse engineering, 3D printing, industrial inspection, robotics, augmented reality (AR), autonomous driving, and other fields [2,3,4]. Traditional depth sensors, such as the Structured Light Camera, Time of Flight (ToF) Camera, and Terrestrial Laser Scanner (TLS), provide depth data as depth images or sets of unorganized points. These sensors have particular tradeoffs in terms of speed, accuracy, flexibility, and price [5]. The hand-held laser line scanner projects a laser pattern onto the surface of the object, and densely measures the surface profiles (3D curves) based on the optical triangulation method. Given its advantages in speed, accuracy, and flexibility [6], this kind of scanner, such as HandySCAN [7], HSCAN [8], and FreeScan [9] have been widely used in high-precision 3D measurement applications in industrial manufacturing.

The scanning process of 3D scanners can be divided into two parts: tracking and mapping [10]. Tracking solves the problem of multiple views registration. These scanners usually use external positioning devices or stable features in the observed scene to estimate the sensor’s position and orientation, which are usually represented by a 6-degree of freedom (DOF) transformation matrix [10]. In the mapping part, we need to align the depth data obtained from different viewpoints into a common reference frame by using the transformation matrix, and then use the depth data to build a model with a given resolution [11]; the mapping process needs to remove redundant data to the resolution of the final model and refine the observed surface continuously, since it is almost impossible to build a complete model without observing some areas twice or more, and the position estimation of the sensor is always inaccurate to some degree [6]. In other words, real-time 3D reconstruction requires incremental fusion of the depth data from different viewpoints into a single 3D representation that is continuously reparametrized [12]. This process is usually referred to as geometric fusion [6].

Fusion approaches based on different principles have been proposed. Point-based fusion methods use simple unstructured representations that closely map to the input depth data through the measured ray to merge overlapping points [13,14,15]. However, these methods cannot generate continuous surfaces directly. Height-map based methods use a more compact 2.5D continuous surface representation [16,17]. Although these methods effectively compress the surface, they fail to reconstruct the complex 3D structure. Implicit volumetric methods have demonstrated convincing results for incremental 3D reconstruction [18,19,20]. The volumetric representation is also referred to as the truncated signed distance field (TSDF), which represents the reconstructed surface by samples of a distance function continuously stored in volumetric grids (voxels) [21]. The reconstruction result can be obtained by extracting the isosurface from the zero-level set of the signed distance function [22,23]. This approach is topologically flexible and has linear complexity. With the help of massively parallel processors, it has become very popular in real-time 3D reconstruction [1,4,19,20,24,25].

In the voxel-based fusion method, Bylow et al. [26] improve the weight function to reduce the impact of unreliable measurements. Dong et al. [27] model the internal probability distribution of SDF values to eliminate outlier observations. Other methods use RGB image to improve the level of detail of surface representation, but they cannot achieve real-time performance [28,29,30].

Some high-precision sensor-tracking methods [26,31,32] using the features of volumetric representation are also proposed. Furthermore, some other research has focused on introducing global adjustment in real-time reconstruction to reduce the tracking drift accumulation caused by frame-to-frame registration, as well as the corresponding voxel data fast update strategy [19,33,34,35,36]. In addition to the static scene reconstruction, voxel-based methods can also be used in the 3D reconstruction of some dynamic scenes, such as reconstructing the human body surface undergoing free non-rigid motion [25,37,38,39,40,41].

Recently, the application of machine-learning methods in voxel-based 3D reconstruction, especially in the form of deep neural networks, has become a new trend [10]. Deep learning has achieved impressive results in processing sensor noise [42], completing scans of scenes [43], and 3D scene semantic segmentation [44,45,46,47,48,49].

Most of the fusion methods require the depth maps or sets of unorganized points as the input. Little research has been based on 3D laser curves. Tubić et al. first proposed a method of constructing TSDF using intersecting curves and a method of the pose refinement of curves by TSDF [6]. This method has been successfully exploited in the hand-held laser line 3D scanning system.

It was mentioned by [21] that TSDF is problematic for reconstructing thin surfaces, but did not provide corresponding solutions. Moreover, no relevant research has been found to address this problem. The influence of the observed surface extends behind their estimated position and interferes with the TSDF from the opposite side of the thin surface. This will cause the surface to shrink towards the interior of the model, and make some topological errors appear on the surface, such as overlap, intersections, or gaps. This effect can be mitigated by reducing truncation, but there is still a set limit for the object’s thickness. The first contribution of this paper is to propose a thin-surface attribution judgment method to distinguish opposite surfaces and store them separately, which effectively avoids the interference of observations of thin surfaces. In some exceptional cases, real-time thin-surface attribution judgment may fail, so we have developed a post-processing method to detect and repair regions that have topological problems.

The efficiency of the underlying data structure is another critical issue in real-time 3D reconstruction. Many advanced real-time reconstruction systems have implemented different underlying data structures on graphics processors, but they have made tradeoffs in terms of speed, quality, and scale. The kinect fusion [18] achieves a real-time frame rate but has a large memory footprint since it uses the volume to represent the whole space instead of only reconstructing densely near the observed surface. The moving volume method uses the motion of a camera to stream data from the graphics processing unit (GPU) to central processing unit (CPU) to expand the space [50]. However, the current active volume structure is the same as the Kinect Fusion, and a lot of memory is required to represent the empty space. In recent years, sparse volumetric representations have been developed to compress space [20]. The main idea is to reconstruct densely only near the observed surface and discard empty space. To index the allocated data, octree [51,52,53], and hash table [1,19,20,24,27,34] have been successfully applied. An octree is unsuitable for graphics processors because of its low degree of parallelism. Nießner et al. proposed a voxel-hashing method for storing TSDF. The complexity is *O* (1) for accessing, inserting, and deleting data in hash tables; this method both effectively reduces the memory footprint and has a high-speed advantage [20].

Hashing-based fusion methods can even run on mobile platforms with relatively low computing capability, such as tablets and mobile phones [24,31,35,54]. Based on the work of Nießner et al., Kahler et al. improved the storage structure of the hash table to reduce the impact of hash conflict [24]. Moreover, they have developed a hierarchical structure based on hash with few levels to improve storage efficiency further [1]. A method called structured skip list also has been proposed to increase the number of voxels that a hash table can accommodate [55]. Although these improved hashing methods alleviate the memory pressure to some degree by compressing data, they do not solve the problem from the root. GPU memory still limits the spatial extent and resolution of the voxel grid. Nießner et al. alleviate the memory pressure of GPU through bidirectional GPU–CPU streaming, but the GPU memory may still be exhausted when reconstruction scenarios are very dense. Our second contribution is to improve the data structure of voxel hashing. We only store hash tables on the GPU, and all voxels are stored on the CPU; thus, the limitation of GPU memory capacity is lifted on the scale of the reconstructed scene.

In this work, aiming at the problem of thin-surface reconstruction in the existing TSDF fusion method, we add a step of thin-surface attribute judgment in the real-time process to eliminate the mutual interference between the opposite sides of the thin surface; the angle between the surface and the observed ray is used to distinguish the TSDF observations from different parts of the thin surface. For the regions where surface attribution misjudgment may occur, we exploit a method of weak topological regions repairing to fix the possible topological errors in these regions. To solve the problem that the underlying data structure of the existing TSDF method takes up too much memory of the GPU, we design a new memory management method to reduce the memory utilization of GPU. By moving all voxel data to CPU, the memory pressure of GPU is alleviated, and the scale of reconstruction scene is further increased.

The remainder of this paper is structured as follows: Section 2.1. outlines the TSDF implementation based on 3D curves. Section 2.2. describes our improvements to thin-surface reconstruction in real-time processing; Section 2.3. explains a post-processing method of repairing weak topological regions, Section 2.4. exposes the underlying data structure for storing TSDF. The results and discussion follow in Section 3. Finally, we conclude and outline some ideas for future work in Section 4.

## 2. Methodology

### 2.1. Truncated Signed Distance Field (TSDF) Computation from 3D Curves

Our real-time 3D reconstruction approach is implemented with a highly integrated hand-held laser scanner, as shown in Figure 1. The scanner measures the 3D curves (a set of 3D points) of the object’s surface using laser triangulation [56]. A laser pattern is projected onto the measured object, and the left and right cameras capture the 2D images of the object at the same time; from the 2D images, we extract the 2D surface curves of the projection pattern reflected on the object; a set of 3D curves is computed in the sensor coordinate system using the 2D surface curves. Meanwhile, the 3D curves measured from different frames are aligned to the global coordinate system by the sensor’s position and orientation, which are continuously computed from the circular mark-points pasted on the object surface. Our 3D reconstruction algorithm integrates these arbitrary intersecting 3D curves into a single volumetric representation and refines the reconstructed surface continuously based on [6].

TSDF constructed from 3D curves is a vector field, which is different from the classical signed distance field [21]. As shown in Figure 2a, each voxel **v** in a field stores the distance vector to its closest point on the reconstructed surface. The vector’s direction is parallel to the normal of the closest point on the surface. The sign of the distance indicates whether a voxel **v** is inside or outside the surface. If **v** is outside of the surface, its sign is positive, and if **v** is inside the surface, its sign is negative. Since voxels that are too far from the surface have little effect on the surface construction, the distance field is truncated at distance |d|≤T. Truncation *T* affects not only the effect of surface reconstruction but also the runtime performance and memory consumption of the algorithm. On the one hand, if *T* is set too large, the surface details will be smoothed out. At the same time, increasing *T* means processing more voxels, thus slowing down the algorithm and consuming more memory. On the other hand, if *T* is smaller than the level of noise, it will lead to the appearance of isolated patches around the reconstructed surface. Therefore, it is generally appropriate to set *T* to the minimum value larger than the noise level of the apparatus, which is 0.9 mm for the scanner used in our experiment. Truncation *T* is also the leading cause of thin-surface problems in TSDF. How *T* affects thin-surface reconstruction will be mentioned in the next section.

Reconstruction of surfaces is based on such a fundamental principle that two intersecting lines can determine a plane. Then the local reconstructed surface patch can be approximated in the neighborhood of the intersection points of all the curves on the surface. Figure 2b illustrates the principle of reconstruction in a simple case where two curves intersect. Assuming that **p**_1_ and **p*_2_*** are the closest points to voxel **v** on curves *c*_1_ and *c*_2_, respectively, **v**_1_ and **v**_2_ are two vectors from the voxel center point **p_v_** to **p**_1_ and **p**_2_, respectively. The tangents of the curves at **p**_1_ and **p*_2_*** have been computed in advance and are expressed as **t**_1_ and **t**_2_, respectively. Since the estimated normal **n** of the tangent plane to the surface should be perpendicular to **t**_1_ and **t**_2_, **n** should be the cross-product of **t**_1_ and **t**_2_. The distance *d* from **p_v_** to the closest surface can be approximated as the average of the projections of **v**_1_ and **v**_2_ on **n**. Therefore, the value **f** of the distance vector field at **v** is *d***n**. A full description of how to locate the closest points on a curve and how to calculate the tangent at the closest point is beyond the scope of this paper. Interested readers can find relevant content in [6].

More formally, let *n* curves of *c*_1_, …, *c*_n_ pass through the local surface patch; **p**_1_, …, **p**_n_ are their respective closest points to voxel **v** and **t**_1_, …, **t**_n_ are the tangents of **p**_1_, …, **p**_n_ respectively. Then the distance from the central point **p_v_** of voxel **v** to the surface $\overset{\wedge }{S}$ can be obtained as the average value of the projected distance vectors on the estimated normal, i.e.,:(1)$$d(v,\overset{\wedge }{S})=\frac{1}{n}\sum_{i=1}^{n}\langle p_{i}-p_{v},n\rangle$$
where **p**_i_ is the closest point to **p_v_** on the curve *c**_i_*, and **n** is the estimated normal of the surface.

Since the normal **n** is perpendicular to all tangents **t**_1_, …, **t**_n_, the following equation can be obtained:(2)$$\frac{1}{n}\sum_{i=1}^{n}t_{i}t_{i}^{T}n=Cn={\left[\begin{array}{ccc} 0 & 0 & 0 \end{array}\right]}^{T}$$

**C** is the covariance matrix of tangents, and the eigenvector associated with the minimum eigenvalue of **C** is the solution to **n**. Then, the value of vector field **f**: ℝ3→ℝ3 at **v** is:(3)f(v)=d(v,S∧)n

Therefore, the closest surface point **p**_c_ of voxel **v** can be obtained by:(4)$$\;p_{c}=p_{v}+f(v),\;p_{c}\in \overset{\wedge }{S}$$

A curve can be considered as a linear interpolation of measure points {**p**_1_, …, **p***_n_*} represented by a set of line segments *c* = {*l*_1_, …, *l_n-1_*}, where *l_i_* = $\bar{p_{i}p_{i+1}}$. In online scanning, the measured surface curves are used to update the covariance matrix **C** and distance *d* of all neighboring voxels incrementally. Each curve only affects voxels within a certain vicinity. As shown in Figure 3, the voxels affected by each line segment $\bar{p_{i}p_{i+1}}$ on a curve can form an approximate cylinder (called a fundamental cell) whose radius is determined by the truncation distance, which is called the envelope size in [6]. The envelope here serves the same purpose as the truncation mentioned earlier. Curves only affect voxels in the envelope and do not affect voxels beyond it. The following low-pass weight function prevents the discontinuity of the field over the envelope boundary:(5)$$\omega \left(d\right)=e^{\frac{-d^{2}}{\sigma^{2}}}$$

*σ* stands for the size of the envelope. Increasing the envelope size can make the reconstructed surface smoother, but it will also lose any sharp features. Conversely, reducing the envelope size makes the surface noisy. Tubić et al. recommended $\frac{\sqrt{3}}{2}\cdot \delta$ as the envelope size as in [6], and δ as the voxel size.

After weighting with *ω_i_*, new observations are integrated into the individual field as follows:(6)$$C=\frac{1}{\sum_{i=1}^{N}\omega_{i}}\sum_{i=1}^{N}\omega_{i}t_{i}t_{i}^{T},$$
(7)$$\tilde{v}=\frac{1}{\sum_{i=1}^{N}\omega_{i}}\sum_{i=1}^{n}\omega_{i}\left(p_{i}-p_{v}\right),$$
(8)$$\omega =\sum_{i=1}^{N}\omega_{i}.$$

The average distance-vector $\tilde{v}$ is projected onto the normal **n** calculated from the covariance matrix **C** to obtain the value of TSDF at **v**:(9)$$f(v)=\frac{\tilde{v}\langle \tilde{v},n\rangle }{{\left\|\tilde{v}\right\|}^{2}}$$

**f**(**v**) is only calculated via real-time rendering and post-processing to generate surface meshes. The reconstructed surface can be obtained by extracting the isosurface from the zero-level set of **f**(**v**). In the online process, raycasting is used to render the reconstructed surface from the current view. After stopping scanning, a marching cubes method can be used to generate mesh models.

The above content briefly introduces method of Tubić et al. [6] of generating a surface model from a set of intersecting 3D curves. This method is topologically flexible and has linear complexity, but it cannot reconstruct the surface of very thin objects. We use a laser line scanner to measure the 3D curves from the surface of a sheet metal part with a thickness of 0.8 mm and use method of Tubić et al. to reconstruct the surface mesh model of this object, which is shown in Figure 4a. The resolution (voxel size) of the model is 0.5 mm, and the size of the truncation (also known as the fundamental envelope) is 0.9 mm. Tubić’s output has a large area of data missing, and its distance (one-sided Hausdorff) to the computer-aided design (CAD) model is greater than 0.2mm (Figure 4b). We optimize the robustness of Tubić’s method to thin objects. With the same resolution and truncation, the optimized results not only guarantee the completeness of data (Figure 4c) but also guarantee its maximum distance (one-sided Hausdorff) to the CAD model within 0.2 mm (Figure 4d). The proposed thin-surface optimization of Tubić’s method is described in Section 2.2 and Section 2.3. Figure 5 shows the whole pipeline of the proposed algorithm. The algorithm employs a hash table to organize the underlying voxel data. The content related to the underlying data structure can be found in Section 2.4. The whole procedure of the proposed 3D reconstruction algorithm can be summarized as the following seven steps:Voxels Allocation: after receiving a frame of 3D curves data, we compute the **p_v_** of all voxels that fall within the fundamental envelopes of the curves and find the corresponding voxel block of **p_v_** in the hash table. If no block is found, we allocate a new block for it on the heap;TSDF Computation: compute the TSDF observations of these voxels (including the closest point **p**_c_ on the curve and tangent **t**_c_ on **p**_c_);Voxels Indexing: use **p_v_** to index the address of voxel **v** in hash table;Voxel Integration: before integrating the new TSDF, it is necessary to determine whether the new one is on the same side as the surface already encoded by **v**. If so, the new TSDF is integrated to voxel **v** by Equations (6)–(8) (updating weight *ω*, matrix **C** and distance vector $\tilde{v}$). Conversely, we find the inverse voxel **v**’ on the other side of the thin surface and update **v**’. The method of determining which side of the thin surface the current observation belongs to will be elaborated in the next section;Normal filtering: the surface normal plays a vital role in the judgment of thin-surface attribution, and also affects the real-time display quality. Therefore, we need to denoise it to make it reliable. Firstly, the surface point **p**_c_ (Equation (4)) and normal **n** (through a singular value decomposition (SVD) of matrix **C**) encoded by voxel **v** are computed. Then the neighborhood points of **v** are found in the hash table, and **n** is recomputed with the surface points and normals of **v**’s neighbors. The related content of normal filtering will also be described in the next section.Weak Topological Regions Repair: although we have distinguished the opposite sides of the observed thin surface when integrating voxels, it is difficult to distinguish them in some areas with complex topology. When the user stops scanning, we adopt a post-processing method to detect and repair the topological errors in these areas automatically.Isosurface Extraction: after repairing the topological errors in TSDF, we use the marching cubes algorithm to extract the isosurface and generate a mesh model.

### 2.2. TSDF for Thin Surface

As mentioned earlier, TSDF has a disadvantage for reconstructing thin surfaces. The distance field extends a band, i.e., truncation, both inside and outside the observed surface. When the surface’s thickness is less than the width of the truncation band, the values of distance fields that encode opposite sides of the surface will interfere with each other, leading to the effect that the rear surface cannot be recovered from the fields. Figure 6a shows a 2D slice of a corrupted thin surface. Black solid lines represent the real surface of a thin object, and each solid arrow indicates a value in the distance field. The truncation distance of the field is two times of voxel size, and the interfered voxels are colored gray; the darker their color, the higher the impact. Unaffected lattices are white because they are too far from the other side of the surface. These arrows of interfered voxels do not point to the real surface, but to multiple false surfaces that are represented by red dotted lines. Therefore, the correct method is to store two vectors (dotted arrows) in a gray lattice, as shown in Figure 6b. That is to say, each voxel in the overlapping region should encode two tangent planes that are on different real surfaces. It is first necessary to determine which side of the thin surface each TSDF observation originates from during online scanning; we call this process thin-surfaces attribution judgment.

When one side of a thin surface is observed, the opposite side can be occluded. The view direction of the current observations implies that the current observations belong to one side of the thin surface. Moreover, to obtain the most accurate observations, the direction of the sensor’s view usually needs to be approximately perpendicular to the object’s surface. Therefore, the correlation between the direction of view and the attribution of the thin surface is stronger. Thus, we store the line of sight of each voxel and continuously update it during scanning as follows:(10)$$\mathrm{sight}\left(v\right)=\{\begin{array}{ll} v-v_{p} & \;\mathrm{if}\;\mathrm{sight}\left(v\right)={\left[\begin{array}{ccc} 0 & 0 & 0 \end{array}\right]}^{T}\\ \mathrm{sight}\left(v\right) & \mathrm{if}\;\left|\langle v-v_{p},n_{v}\rangle \right|\leq \left|\langle \mathrm{sight}\left(v\right),n_{v}\rangle \right|\\ v-v_{p} & \mathrm{if}\;\left|\langle v-v_{p},n_{v}\rangle \right|>\left|\langle \mathrm{sight}\left(v\right),n_{v}\rangle \right| \end{array}.$$
here, **n_v_** denotes the normal of the surface encoded by voxel **v**, and **v**_p_ represents the current viewpoint. Each normal **n_v_** computed from the covariance matrix **C** in Equation (6) must be oriented consistently towards the viewpoint **v**_p_ (so that the angle between **n_v_** and the line of sight is less than $\frac{\pi }{2}$). Equation (10) guarantees that the angle between the line of sight and the surface normal is the smallest so that the sign of the normal **n_v_** determined by the line of sight is reliable.

In the online scanning process, there are two cases of the viewpoint orientation:One Side Visible: when the line of sight is oriented toward one side of the thin surface, the sensor can only measure the current side and cannot measure the opposite side because the dominant side occludes it.Both Sides Visible: when the line of sight is approximately parallel to a thin surface and perpendicular to the edge of a thin surface, the thin surface on the opposite sides near the edge can be observed.

The first case (One Side Visible) must be addressed to prevent the thin surface from being totally corrupted during online scanning. The second case (Both Sides Visible) will only affect the reconstruction quality near the edge of the thin surface. Moreover, it requires a more complex mechanism, and its computation cost is very high. Therefore, we put this in the post-processing process, whose specific details are described in Section 2.3.

The signs of TSDF indicate whether a voxel is inside or outside an object. We use the following procedure to determine the signs of TSDF. Firstly, the orientation of the normal **n_v_** is adjusted to be consistent with the distance vector $\tilde{v}$ (Equation (7)) to get the vector **n***_dis_*, that is, **n***_dis_* needs to satisfy the equation:(11)$$\;n_{dis}\cdot \tilde{v}>0$$

Then the sign *S***_v_** at voxel **v** is determined by the dot product of **n***_dis_* and the line of sight **V_v_** stored in **v**:(12)$$\mathrm{sign}\left(v\right)=\{\begin{array}{cc} 1 & \mathrm{if}\;\langle n_{dis},V_{v}\rangle \leq 0\\ -1 & \mathrm{if}\;\langle n_{dis},V_{v}\rangle >0 \end{array}$$

With the change of signs, we can quickly judge the surface attribution online. When scanning from one side of a thin surface to the other, the current direction of view ${V^{\prime }}_{v}$ will deviate significantly from the previous **V_v_** stored in voxel **v**. If the normal ${n^{\prime }}_{v}$ is oriented to the current viewpoint, the sign ${S^{\prime }}_{v}$ determined by ${n^{\prime }}_{v}$ and ${V^{\prime }}_{v}$ is contrary to the sign *S***_v_** previously stored in **v**. Meanwhile, if the signs ${S^{\prime }}_{v}$ and *S***_v_** are the same, the current measurement is on the same side as the previous cumulative measurement. If the current observation is judged to belong to the other side of the thin surface, then a new voxel $v^{\prime }$ is allocated on the heap, the observed value is integrated into $v^{\prime }$, and the address of the new voxel $v^{\prime }$ is stored in the original voxel **v**. When querying the value of voxels in TSDF, we need to know not only the coordinates of the query point but also the line-of-sight at this moment to determine whether the voxels we are looking for are **v** or **v**’. Section 2.4 provides more details about the data structure.

Figure 7 depicts the plot of the sign *S* varying with the angle *α* between vector **n***_dis_* and line of sight **V_v_**. As shown in Figure 7, near the critical point ($\alpha =\frac{\pi }{2}$), a slight change in the angle will cause the sign *S* to be flipped. Small perturbations of data (related to the accuracy of apparatus) will lead to a certain degree of deviation in normal estimation. When the angle between the line of sight and the surface normal $\alpha \approx \frac{\pi }{2}$, the error of normal estimation will lead to an inaccurate sign of TSDF, which will make the misjudgment of the thin-surface attribution. Meanwhile, the reliability of the measures obtained in this case is also relatively low. Therefore, a current measurement reliability judgment is added to the process to determine whether the thin-surface attribution judgment can be turned on. When the line of sight and the surface normal are almost perpendicular ($\left|\langle n_{dis},V_{v}\rangle \right|\leq 0.05$), the current measurement is judged to be unreliable and discarded. Conversely, the thin-surface attribution judgment is turned on, and the current observation is integrated into the corresponding voxel determined by the thin-surface attribution judgment.

From the above description, we can see that the accuracy of a normal estimation will affect the thin-surface attribution judgment and the accuracy of the distance vector field **f**(**v**). Figure 8 shows the surface points (Equation (4)) and normal (through a SVD decomposition of matrix C in Equation (6)) of a scanned model. As can be seen in Figure 8a, the normals are very noisy and unreliable. A normal filtering method with lower computational complexity is used in the online scanning process to improve the accuracy of normal estimation. Figure 8b shows the filtered results. The overall flow of online normal filtering is:Perform the approach proposed in [6] to estimate the first normal **n***_f_* and compute the reliability measure λ*_r_* of **n***_f_*;For each voxel **v** in the field, estimate the normal **n***_s_* based on an anisotropic neighborhood;Update the normals by combining the first normal **n***_f_* with **n***_s_* under the influence of coefficient λ*_r_* until either convergence or a maximum number of iterations is reached.

When estimating the first normal **n**_f_ of voxel **v**, the three eigenvalues λ_1_ > λ_2_ > λ_3_ of the covariance matrix **C** in Equation (6) are used to evaluate the reliability coefficient λ of **n**_f_ Let
$$\lambda_{\sigma }=\frac{\lambda_{1}}{\lambda_{1}+\lambda_{2}+\lambda_{3}},$$

Pauly et al. [57] defined this coefficient as the surface variation. This measures the ratio of variance in the normal direction to the total variance and can be used to express the reliability of normal estimation. This quantity ranges between 0 (flat case) and 1/3 (anisotropic case). Add the following weight function to λ*_σ_* so that the range of coefficient λ*_r_* is [0, 1]:(13)$$\lambda_{r}(\lambda_{\sigma })=\{\begin{array}{ll} 1 & \mathrm{if}\lambda_{\sigma }<\rho \\ -\varphi \ln(\lambda_{\sigma })-\psi & \mathrm{if}\lambda_{\sigma }\geq \rho ,\;\mathrm{and}\;-\varphi \ln(\lambda_{\sigma })-\psi \geq 0\\ 0 & \mathrm{if}-\varphi \ln(\lambda_{\sigma })-\psi <0 \end{array}$$

The parameters *ρ*, *φ*, and *ψ* can be determined experimentally. Here, we use *ρ =* 0.01, *φ =* 0.1, *ψ =* 0.15. The purpose of introducing coefficient λ*_r_* is to preserve the features of the first normals as much as possible and to prevent over-smoothing.

Let **v** be the voxel at which we need to re-estimate the normal; **p** represents the point closest to **v** on the surface, and **n** represents the first normal. **p***_i_* is the surface point at voxel **v***_i_* ($v_{i}\in N_{v}$) in the neighborhood of voxel **v**, and **n***_i_* is first normal at **v***_i_*. $N_{v}$ is under a neighborhood size *σ*_p_ specified by users. A covariance matrix **C_n_** can be constructed from all normals **n***_i_* in the neighborhood of voxel **v** as follows:(14)$$C_{n}=\frac{1}{\sum_{i=1}^{N}\omega_{i}}\sum_{i=1}^{N}\omega_{i}\left(n_{i}-n_{m}\right)\cdot {\left(n_{i}-n_{m}\right)}^{T}$$
here, **n***_m_* is the average of **n***_i_*. The weight *ω* consists of two factors:$$\omega =\omega_{r}\cdot \omega_{\theta },$$
where *ω_r_* is used to evaluate the reliability of neighborhood point **v***_i_* equal to the weight (Equation (8)) accumulated at **v***_i_*. Therefore, *ω_r_* is related to the number of observations at voxel **v***_i_* and the distance from voxel **v***_i_* to the surface. The weights are more significant when there are more observations, or the distance is closer, whereas the weights are smaller otherwise.

*ω_θ_* is an anisotropic weight function, as in the bilateral normal filtering method [58]; it is defined as follows:(15)$$\omega (p_{i},n_{i})=e^{-{\left(p_{m}-p_{i}\right)}^{2}/\sigma_{p}^{2}}\cdot e^{-{\left(\frac{1-n_{m}^{T}\cdot n_{i}}{1-\cos(\sigma_{n})}\right)}^{2}}$$
where *σ***_p_** is the size of the neighborhood, and the angle parameter *σ***_n_** measures the similarity of the neighborhood normals. **p***_m_* and **n***_m_* are the average of **p***_i_* and **n***_i_*, respectively. The eigenvector associated with the minimum eigenvalue of **C_n_** is the new estimate **n**_s_ at voxel **v**, since it corresponds to the direction of the smallest variance of the neighborhood normal.

The final normal estimate is updated iteratively by the following formula:(16)$$n\propto \lambda_{r}n_{f}+\left(1-\lambda_{r}\right)n_{s}$$
where ∝ denotes the normalization of the length such that **n** is a unit vector. Whenever voxel **v** gets a new **n**_*fi*_, a new normal estimate **n***_i_* is obtained by the above method until the deviation between **n***_i_* and **n***_i_*_−1_ is less than a certain threshold, or the number of updated times reaches a limit. As a local filtering method, our method has a low computational cost. Its time-consuming is related to the neighborhood’s size; a large neighborhood may increase the time consumption but will also lead to over-smoothing. The 3 × 3 × 3 neighborhood proved sufficient.

### 2.3. Repairing Weak Topological Regions

As mentioned in the previous section, when the line-of-sight is almost parallel to the thin surface, the thin-surface attribution judgment may fail, so the distance vector in the vector field cannot point to the correct position. The reconstructed surface mesh data from the vector field will deviate significantly from the object’s actual surface and will result in topological errors such as overlap, intersections, or gaps. Without prior knowledge, it is impossible to get the desired surface shape by merely repairing the topological errors in the surface mesh. Therefore, the distance vectors in TSDF must be repaired to restore the locus of the real surface. We need to analyze TSDF to find areas that may have topological problems and then repair the values of these areas.

#### 2.3.1. Detecting Weak Topological Regions

Our method of detecting weak topological regions in TSDF is inspired by Sharf’s method [59]. Sharf’s approach for repairing weak topological regions is to find the critical point of the signed distance function (that is the point where the gradient disappears), and then specify the locus of the correct surface at the critical point by user interaction. Sharf finds discrete critical points, not contiguous regions. Many points with topological errors may still be undetected near these critical points, but Sharf compensates for this problem through specific user interface interactions, which violates our original intention of automatically detecting and repairing weak topological areas. Here, we make some adjustments to Sharf’s method so that the weak areas we find are continuous and complete. Specifically, the methods of detecting weak regions can be summarized as follows:
Find the critical points: find all voxels $v_{i}\in N_{v}$ in a neighborhood with **v** as the center. **v***_i_* is divided into two sets, one *f*(**v***_i_*) > *f*(**v**) is defined as positive, and one *f*(**v***_i_*) ≤ *f*(**v**) is defined as negative. Within each set, adjacent voxels are joined together to obtain several connected groups ***Γ****_i_^±^*, as shown in Figure 9. If the number of connected groups of **v** is 2, then **v** is not the critical point. If the number of connection groups of **v** is 1 or ≥ 2, then **v** is a critical point;Generate weak topological regions: take each critical point **P**c as a seed point, find connected component **P**g (a region of adjacent voxels) of **P**c within a specified range (such as two times of the voxel size), and the dot product of the normal less than a certain threshold *T_dot_*. Figure 10 shows weak regions detected by our method. This model has many stiffeners with a thickness of about 1 mm. When the truncation is also set to 1 mm, the surface model is prone to generate topological errors in the location of these stiffeners. The proposed method can detect these weak topological regions (the red areas in the figure), and the method of repairing topology errors will be described in the next section.


#### 2.3.2. Repairing Weak Topological Regions

Figure 10b shows that some points in the weak region are unaffected by the thin-surface effect and can fall on the actual surface. By a reasonable parameter configuration (including voxel size and truncation width), we can ensure that there will always be some reliable points in the weak region. Therefore, the task of repairing weak areas can be transformed into the process of finding reliable points and inferring the value of unreliable points from reliable points.

As shown in Figure 11, voxels inside convex surfaces are easily affected by observations on both sides of the thin wall, while voxels outside are unaffected. Therefore, for convex surfaces, the TSDF values of voxels with positive signs can be used as input to infer the values of voxels with negative signs. Conversely, for concave surfaces, the TSDF values of positive voxels can be inferred from negative voxels. The number of positive and negative connection groups in weak regions can be used to judge whether a surface is convex or concave. When the number of positive connection groups is larger than that of the negative connection groups, the surface is convex, whereas the surface is concave otherwise.

After identifying reliable voxels, we need to infer the actual surface from them. **p***_i_* and **n***_i_* are the surface points and normals of reliable voxels, respectively. Our goal is to construct a directed distance field f:ℝ3→ℝ so that *f* (**p***_i_*) = 0 and ∇f(ni)=ni. The surface represented by *f* passes through all **p***_i_*, and the normal of the implicit surface matches **n***_i_* at **p***_i_*. In calculating *f*, we use the Hermite radial basis function (HRBF) [60] interpolation method since it considers both points and normals. Then, we define *f* as follows:(17)f(x)=∑iNαiϕ(‖x−pi‖)+∇[ϕ(‖x−pi‖)]Tβi
where **x**∈ℝ3 is used to indicate the location to be evaluated. Function *ϕ*: ℝ→ℝ is a real-value function on [0, ∞) called the radial basis function. When fitting functions with three variables, biharmonic (*ϕ*(r) = r) and triharmonic (*ϕ*(r) = r^3^) splines are good choices.

∇f(x) is the column vector of dimension *d = 3*, and can be calculated by the following formula:(18)$$\nabla f(x)=\sum_{i}^{N}\alpha_{i}a_{i}\left(x\right)+\left(B_{i}(x)+C_{i}(x)\right)\beta_{i}.$$
where $a_{i}(x)=\phi^{\prime }\left(\left\|x-p_{i}\right\|\right)\frac{x-p_{i}}{\left\|x-p_{i}\right\|}$,

$B_{i}(x)=\left(\begin{array}{ccc} b_{i}(x) & 0 & 0\\ 0 & \ddots & 0\\ 0 & 0 & b_{i}(x) \end{array}\right)$ is a diagonal matrix of dimension *d* × *d* with $b_{i}(x)=\frac{\phi^{\prime }\left(\left\|x-p_{i}\right\|\right)}{\left\|x-p_{i}\right\|}$,


$\begin{array}{l} C_{i}(x)=c_{i}(x)\cdot \left(\left(x-p_{i}\right)⊗\left(x-p_{i}\right)\right),\mathrm{with}\\ c_{i}(x)=\frac{1}{{\left(\left\|x-p_{i}\right\|\right)}^{2}}\left(\phi^{\prime \prime }\left(\left\|x-p_{i}\right\|\right)-\frac{\phi^{\prime }\left(\left\|x-p_{i}\right\|\right)}{\left\|x-p_{i}\right\|}\right). \end{array}$


To calculate the value of *f*, we must find *N α_i_*∈ℝ and **β***_i_*∈ℝ3. According to the constraints of *f* (**p***_i_*) = 0 and ∇f(ni)=ni, the following linear equations are obtained:(19)(f(pi)∇f(pi))=(cniT)

Therefore, the equation system **Ax**^T^ = **b**^T^ can be written as:(20)$$\left(\begin{array}{ccc} f_{1}^{T}(x_{1}) & \cdots & f_{N}^{T}(x_{1})\\ \nabla f_{1}(x_{1}) & \cdots & \nabla f_{N}(x_{1})\\ \vdots & \ddots & \vdots \\ f_{1}^{T}(x_{N}) & \cdots & f_{N}^{T}(x_{N})\\ \nabla f_{1}(x_{N}) & \cdots & \nabla f_{N}(x_{N}) \end{array}\right)\left(\begin{array}{c} \alpha_{1}\\ \beta_{1}\\ \vdots \\ \alpha_{N}\\ \beta_{N} \end{array}\right)=\left(\begin{array}{c} 0\\ n_{1}\\ \vdots \\ 0\\ n_{N} \end{array}\right)$$

A single matrix block *(d + 1) × (d + 1)* of coefficient **A**
*_(N(d+1))×(N(d+1))_* can be written as:(21)$$\left(\begin{array}{c} f_{i}^{T}(x_{j})\\ \nabla f_{i}(x_{j}) \end{array}\right)=\left(\begin{array}{cccc} \phi \left(\left\|x_{j}-p_{i}\right\|\right) & \frac{\phi^{\prime }\left(\left\|x_{j}-p_{i}\right\|\right)}{\left\|x_{j}-p_{i}\right\|}\left(x_{j,x_{1}}-p_{i,x_{1}}\right) & \cdots & \frac{\phi^{\prime }\left(\left\|x_{j}-p_{i}\right\|\right)}{\left\|x_{j}-p_{i}\right\|}\left(x_{j,x_{d}}-p_{i,x_{d}}\right)\\ \phi^{\prime }\left(\left\|x_{j}-p_{i}\right\|\right)\frac{x_{j,x_{1}}-p_{i,x_{1}}}{\phi \left(\left\|x_{j}-p_{i}\right\|\right)} & \left(x_{j,x_{1}}-p_{i,x_{1}}\right)\left(x_{j,x_{1}}-p_{i,x_{1}}\right)c_{i}(x_{j})+b_{i}(x_{j}) & \cdots & \left(x_{j,x_{d}}-p_{i,x_{d}}\right)\left(x_{j,x_{1}}-p_{i,x_{1}}\right)c_{i}(x_{j})\\ \vdots & \vdots & \ddots & \vdots \\ \phi^{\prime }\left(\left\|x_{j}-p_{i}\right\|\right)\frac{x_{j,x_{d}}-p_{i,x_{d}}}{\phi \left(\left\|x_{j}-p_{i}\right\|\right)} & \left(x_{j,x_{1}}-p_{i,x_{1}}\right)\left(x_{j,x_{d}}-p_{i,x_{d}}\right)c_{i}(x_{j}) & \cdots & \left(x_{j,x_{d}}-p_{i,x_{d}}\right)\left(x_{j,x_{d}}-p_{i,x_{d}}\right)c_{i}(x_{j})+b_{i}(x_{j}) \end{array}\right)$$

After calculating the weights of *α_i_* and **β***_i_*, the distance from any spatial position **x** to the implicit surface can be calculated using Formula (17), and the gradient can be calculated by Formula (18). The normal of the implicit surface encoded by **x** can be obtained by normalizing the gradient. Therefore, the distance vector and normal of every unreliable voxel in the weak region of topology can be recomputed from the HRBF.

### 2.4. Data Structure

We defined the following structure to store the TSDF properties at each voxel:
 struct Voxel {-float sight [3];-float C [6];-float distance [3];-float weight;-unsigned int pointer;-unsigned int address; };

Each voxel stores the sight, the covariance matrix **C** defined in Equation (6) (Since matrix **C** is a symmetric matrix, only six independent elements in the corresponding upper triangular matrix are stored), the distance defined in Equation (7), the accumulated weight defined in Equation (8), the pointer to the voxel belongs to the other side of the thin surface and the address of the surface point. The normal is stored in the surface point, which can be used to judge the surface attribution and interactive visualization. A certain number of voxels (the default is 8^3^) are merged to store, which is called a block. All voxels in a block are stored continuously in the order of x–y–z, and the coordinates of each voxel center can be obtained from the coordinates of the block.

All voxel blocks are organized by a hash table implemented on GPU [20]. The hash table stores hash entities, each of which points to an allocated voxel block. Their corresponding relationship is established by converting the spatial coordinates into hash values through a hash function. Nießner provided a complete method for searching, inserting, and deleting data in hash tables and a mechanism for resolving hash collisions. Since the hash table is implemented on the GPU, it can index all allocated data in parallel and thus has a considerable speed advantage.

Unlike Nießner, which stores some voxel blocks on the GPU and others on the CPU, we directly store all voxel blocks on the CPU to reduce the memory footprint of the GPU. Only hash tables are stored on the GPU, and each entity in the hash table contains a pointer to the voxel blocks allocated on the CPU. Once we have inserted a new entry into the hash table, we allocate a voxel block to the heap memory on the CPU. The heap is a dynamic 2D array that supports dynamic scaling. It is divided into contiguous blocks (mapped to the size of voxel blocks). Once allocated heap memory is exhausted, a fixed-length array is appended to the back of the 2D array. Since the array is accessed in parallel, synchronization is necessary, by incrementing the pointer of the array using atomic operation.

Before integrating the new TSDFs, all voxel blocks must be allocated that fall on the input 3D curves, and are also within the fundamental envelope of the curves. For each line segment $\bar{p_{i}p_{i+1}}$ on the curve, we first calculate the bounding box determined by $\bar{p_{i}p_{i+1}}$, and then for all the voxels in the bounding box, we determine whether it falls into the fundamental envelope of $\bar{p_{i}p_{i+1}}$. If it is so, we find the corresponding entity of it in the hash table. If no entity is found, we insert an entity in the hash table and allocate a voxel block on the heap.

In addition to the voxel block array, the CPU also stores an inverse voxel block array and an array of neighborhood lookup tables, which are also maintained by dynamic 2D arrays. Inverse voxel blocks are used to store the opposite sides of a thin surface. We also store each element in the inverse voxel blocks as a Voxel struct. Before integrating each new TSDF, we need to determine whether it belongs to the same side of the surface that has been encoded in the non-inverse voxel **v**. If it is so, we integrate the new value to **v**, and vice versa integrate it to the inverted voxel **v**’ pointed by **v**.pointer. If the value of **v**.pointer is empty, a new voxel **v**’ is allocated on the heap of the inverse voxel blocks, and the address of **v**’ is assigned to **v**.pointer. Then we compute the new TSDF (the closet point **p**_c_ on the curve and tangent **t** at **p**_c_) and integrate the new value to its corresponding voxel. When assigning a new inverse voxel v’, we lock voxel **v** atomically to prevent other threads from writing to it at the same time.
**Algorithm 1.** Computation of the TSDF from a set of curves.**Input:** Set of curves *c_i_* = {**p***_i,_*_1_, …, **p***_i,n_**_i_*}, *i*∈[1,*N*]; tangent **t***_i,j_* at each point **p***_i,j_*, *j*∈[1,*n*].**Output**: **TSDF**: ℝ3→ℝ3.
1: Initialize matrix **C**, distance vector v˜ and weight *ω* in all voxels to zero
2: **for** each curve *i*∈[1, *N*] (number of curves) **do**
3:    **for** each point *j*∈[1, *n*] (number of points on the curves) **do**
4:    Compute the bounding box of the line segment $\bar{p_{i,j}p_{i,j+1}}$;
5:    **for** all voxels **p_v_** (the central point of the voxel) in the bounding box **do**
6:    **if p_v_** is in the fundamental envelope associated with $\bar{p_{i,j}p_{i,j+1}}$ then
7:      Compute the closest point **p***_c_* to **p_v_** on the curve;
8:      Compute the tangent **t***_c_* on the point **p***_c_*;
9:      Find the voxel **v** associated **p_v_** in the hash table;
10:       Find 26 neighborhoods of the voxel block where **v** resides;
11:       Update the elements corresponding to **v’s** block in the neighborhood lookup table;
12:       **if** current TSDF observations (**p***_c_* and **t***_c_*) belong to the surface encoded by **v then**
13:       Update matrix **C** at voxel **v** (Equation (6));
14:       Update the sum of distance vectors $\tilde{v}$ with **p***_c_*–**p_v_** (Equation (7));
15:       Update the sum of weight *ω* at voxel **v** (Equation (8));
16:       Compute the surface point **p***_s_* (Equation (4)) and normal **n** (by SVD decomposition of
17:       matrix **C**);
18:       Find the neighborhoods of **v**;
19:       Recompute **n** (Equation (16)) with the normals of the neighborhood points;
20:       **else**
21:       Find inverse voxel **v**’;
22:       Update matrix **C** at voxel **v**’ (Equation (6));
23:       Update the sum of distance vectors $\tilde{v}$ with **p***_c_*–**p_v_** (Equation (7));
24:       Update the sum of weight *ω* at voxel **v**’ (Equation (8));
25:       Compute the surface point **p***_s_*’ (Equation (4)) and normal **n’** (by SVD decomposition of
26:       matrix **C**);
27:       Find the neighborhoods of **v**’;
28:       Recompute **n**’ (Equation (16)) with the normals of the neighborhood points.

To search voxels neighborhood information conveniently and avoid frequent data transmission between CPU and GPU, we maintain a neighborhood lookup table of all blocks on CPU. Neighborhood lookup table plays a vital role in normal filtering operation. It stores the pointers of 26 neighborhood voxel blocks for all voxel blocks. For each voxel **v**, if it is located at the center of the voxel block **B**, the 26 neighboring voxels adjacent to it can be found in the block **B**; its neighbors can be found by simple subscript addition and subtraction since the elements in a voxel block are stored continuously. If it is on the edge of the voxel block **B**, the neighboring voxels need to be found in the voxel blocks adjacent to **B**. Neighborhood lookup tables are divided into contiguous buckets, each consisting of 26 elements that store pointers to the neighborhood voxel blocks. The storage order of these buckets is the same as that of the voxel blocks. Therefore, all neighbors of each voxel block can be found through its pointer. Neighborhood lookup tables are also updated in real time during scanning. First, we find the voxel blocks in the fundamental envelopes of the current input frame, and then find 26 neighbors for each of them by the hash table to update the corresponding elements of the lookup table. The pseudocode Algorithm 1 shows the whole procedure of computing TSDF based on the above data structure. It is assumed that the tangent of each point on the curve has been calculated before field computation. Figure 12 shows our entire data structure.

## 3. Results

Here, we verify the performance of our proposed algorithm through a series of experiments. The experimental data are all registered 3D curves collected by the laser scanner shown in Figure 1. Figure 13 shows an example of the input data of our algorithm. We use the proposed method to integrate all 3D curves of real objects into TSDF, extract the isosurface, and construct the mesh model.

To verify the effectiveness of the proposed thin-surface improvement, we compare the reconstruction results of the proposed method with Tubić et al. [6] for industrial products of different thicknesses. The accuracy of the proposed method is evaluated by comparing it with two existing hand-held 3D laser line scanners: Creaform HandySCAN700 [7] and Scantech HSCAN771 [8]. As shown in Table 1, these two kinds of scanner can achieve a volumetric accuracy of 0.02 mm (part size within 1 m) and have been widely used in the field of industrial 3D scanning. We use cloud to mesh (C2M) distance metric to quantify the accuracy of surface reconstruction [61]. The distance from each vertex in the source model to the closest triangle in the reference model is recorded, and two standard statistics are computed for all distances: Mean and Standard Deviation (Std.) [61]. In our experiments, the CAD model is used as a reference model. We do statistical hypothesis tests (Welch’s *t*-test [62] and *F*-test [63]) on these two statistics to evaluate the significance of the improvement. All the statistical tests are implemented by R programming language version 3.0.0 [64].

In terms of computational efficiency, we test the speed of the proposed algorithm on three hardware devices with different configurations shown in Table 2. The performance of the devices we used is equivalent to the hardware configuration recommended by mainstream industrial 3D scanner manufacturers (Creaform) [7]. Furthermore, we verify the effectiveness of the proposed method to improve the underlying data structure by comparing its GPU memory usage with VoxelHashing [20].

### 3.1. Experimental Object and Parameter Setting

Figure 14 shows the 3D model of six different objects scanned by the proposed method in our experiment. Figure 14a–e are five thin industrial products with different thicknesses, which are used to verify the effectiveness of the proposed thin-surface optimization. The thickness of these objects is indicated in each subfigure. All of the 3D models in the thin-surface verification experiment are generated with the same configuration parameters: voxel size = 0.5 mm, truncation width = 0.9 mm. For comparison purposes, we also set the same parameters for Tubić et al. [6]. When using the scanners of Creaform [7] and Scannech [8], we also set the resolution of scan to 0.5 mm.

The “relief” (**f**) is 2.4 m × 1.6 m in size. The total number of curves acquired from the “relief” is 472,612, and the total number of 3D surface points is 3,365,470,052. Since this object has large size and low self-occlusion, we use it to test the algorithm time and memory efficiency. This object can ensure that around 7000 laser points are obtained at each frame, basically reaching the upper limit of the single-frame data of the apparatus we used. In the performance test, we compare the speed and memory footprint of the algorithm in different resolutions. The truncation width is set to $\frac{\sqrt{3}}{2}\cdot \delta$ (δ as the voxel size), which is recommended by Tubić et al. [6].

### 3.2. Qualitative Results

Figure 15 compares the surface reconstruction results of Tubić et al. [6] with ours for the same thin objects. The front and back views of each model are shown in these figures, and the front and back sides of the surface model are rendered in different colors (the front is given blue, the back is made yellow). The apparent improvements in the figures are marked with red ellipses. From these figures, we can see that many parts of results of Tubić et al. [6] have been lost, while our results are complete and smooth.

We also used Creaform HandySCAN700 [7] to scan the “sheet metal part” with a resolution of 0.5 mm, as shown in Figure 16. There are some small spikes and holes in Creaform’s [7] model, which are not found in the scanning process. The same defects were not found on the surface of our model. However, compared with Creaform’s [7] results, our surface is not smooth enough on the edges. Alias artifacts appear at sharp features on the extracted surfaces since TSDF is sampled on a regular grid with a fixed step length. In contrast, Creaform [7] uses higher resolution at sharp features.

The “plastic shell” is a more extreme example with a very complex topology. When scanning this object, it is almost impossible to ensure that the line of sight can be perpendicular to the sheets, so almost all of the sheets are corrupted (Figure 17a). In Figure 17b, Creaform’s [7] scanning results show similar problems, and the model surface at the edge of the sheet is messy and incomplete. By repairing the topological structure of these weak topological regions, the shape of these sheets is well restored (the missing parts are caused by occlusion, not by our method).

### 3.3. Quantitative Results

In the accuracy evaluation, we choose two typical thin objects: sheet metal part and plastic shell. The topological structure of the sheet metal part is simple, so it is used to verify the effectiveness of thin-surface improvement (thin-surface attribution judgment) in real-time reconstruction. The plastic shell has a complex topology, which makes it easy to misjudge the attribution of a thin surface in the scanning process; so we use it to verify the effectiveness of the post-processing method for weak topological region repairing.

The accuracy of surface reconstruction is evaluated by the C2M distances [61] between reconstructions and CAD models. Figure 18 plots color maps of C2M distances for the two thin objects. In each subfigure, we visualize the C2M errors through a jet color map and a histogram. Distances greater than 0.2 mm are depicted with deeper colors, and the red/blue gradations indicate positive/negative distances, respectively. We evaluate the C2M errors quantitatively for each method with two statistics, as shown in Table 3. Our real-time processing approach is more accurate than those of Tubić et al. [6], but is similar to the existing scanners (Creaform, Levi, QC, Canada [7] and Scannech [8]). After the proposed post-processing, we get the minimum values on both Mean and Std.

We randomly selected 1000 samples from the C2M distance data population of each method and carried out statistical tests on the samples to evaluate the significance of the proposed improvement. The mean and Std. of each group of sample data are listed in Table 4. Normal distribution is the basic hypothesis of a statistical test, so we use the Pearson’s $\chi^{2}$ test [65] to judge whether the samples are well-modeled by a normal distribution. As shown in Table 5, the *p*-value of all samples is higher than 0.05, so we conclude that the samples are consistent with the normal distribution at the significance level of *α* = 0.05.

Here we apply Welch’s *t*-test (unequal variances *t*-test) [62] to compare the significance of the differences in the mean values between groups of samples, since the Std. between groups is different. Table 6 and Table 7 show the *t*-test results of compared methods for two thin objects. Let ***μ****_i_* be the mean of the population corresponding to a column in the table, and ***μ****_j_* be the mean of the population corresponding to a row. We applied a one-tailed test [66] to each grid in the table at a significance level of 0.05. That is, ***H****_0_*: ***μ****_i_* = ***μ****_j_*, ***H****_1_*: ***μ****_i_* > ***μ****_j_*. If *p* < 0.05, we reject the null hypothesis and conclude that ***μ****_i_* and ***μ****_j_* have a significant difference. For the two thin objects we selected, all the methods involved in the comparison have smaller mean values than Tubić et al. [6], but the differences between the other methods are not significant.

For the difference of Std. between each group of samples, we conduct the *F*-test of equality of variances [63], as shown in Table 8 and Table 9. Let ***σ***_i_^2^ be the variance of the population corresponding to a column in the table, and ***σ***_j_^2^ be the variance of the population corresponding to a row. For each grid in the table, ***H****_0_*: ***σ***_i_^2^ = ***σ***_j_^2^, ***H****_1_*: ***σ***_i_^2^ > ***σ***_j_^2^. If *p* < 0.05, we reject the null hypothesis and conclude that ***σ***_i_^2^ and ***σ***_j_^2^ have a significant difference. For the sheet metal part, the variance of each group can be sorted as follows: Tubić et al. [6] > Creaform [7] > Scantech [8] > Our approach; there is no significant difference between our real-time approach and post-processing approach at the significance level of 0.05. For the plastic shell, Tubić et al. [6] > Creaform [7] > Our real-time approach > Scantech [8] > Our post-processing approach.

### 3.4. Performance

We have implemented various Single Instruction Multiple Data (SIMD) [67] instruction set versions of the proposed algorithm in C++ language, and the program was compiled with Visual Studio 2015. Figure 19a compares the frame rates of different instruction set versions for the online proposed reconstruction method on different hardware devices shown in Table 2. The voxel resolution of scanning is set to 0.5 mm, and about 40,000 voxels are inserted in every single frame. Although on different devices, the algorithm uses CPU multithreading and GPU parallelism to speed up, the use of SIMD instruction set still leads to a frame rate increase. Our algorithm can achieve a real-time frame rate above 60 FPS since most of the CPUs support instruction sets above SSE now. Figure 19b compares the time per frame for our method with Tubić et al. Although our method has additional operations such as thin-surface attribution judgment and normal filtering, the time-consumption increase is not obvious due to the superior performance of the underlying data structure we adopted. The comparison was made on a device with Configuration A.

Figure 20 compares the CPU memory usage and GPU memory usage of our proposed method and VoxelHashing [20] method. The comparison was made on a device with Configuration A. When the voxel resolution was increased to 2 mm, VoxelHashing [20] had already filled up all of the 4 GB GPU memory and made the program crashed, which significantly limited the algorithm’s practicability since most of the GPU hardware devices on the market only supported 4 GB memory. In contrast, the GPU memory footprint of our proposed method was insignificant compared to voxel hashing [20]. Although the CPU memory footprint had increased, it was acceptable (up to 6 GB). After all, most computer devices support CPU memory expansion.

### 3.5. Discussion

From the qualitative results shown in Figure 15, we can see that our proposed modifications have visible progress in the reconstruction of thin surfaces compared with the original method of Tubić et al. [6]. The surface reconstructed by our method is complete and smooth, and it is difficult to see the interference between the opposite sides of the thin surface. However, Tubić et al. [6] cannot achieve the desired results. Both methods use the same input data and configuration parameters (voxel size and truncation width). It can be seen from Figure 17 that the proposed post-processing method is useful and necessary for thin objects with complex topology.

Quantitative analysis results can confirm the conclusion of qualitative evaluation. Whether it is a thin object with simple topological structure (sheet metal part) or complex topological structure (plastic shell), the deviation between the reconstruction of our method and CAD model is the smallest (the minimum mean and Std.). Taking the mean as the evaluation metric, our method significantly improves the accuracy of the original method of Tubić et al. [6], but it shows similar accuracy to the other two existing scanners (Creaform [7] and Scantech [8]). If Std. is used as the evaluation metric, our method is significantly different from other existing methods (Tubić et al. [6], Creaform [7], and Scantech [8]). Our method is more stable than the existing approaches. Moreover, for the thin object with complex topological structure, it can significantly improve the reconstruction accuracy by the proposed post-processing method.

As can be seen from Figure 19, the proposed algorithm can achieve a frame rate of more than 60 FPS on the hardware devices used in the experiment. These devices are not expensive, and their performance is the same as the hardware configuration recommended by mainstream industrial 3D scanner manufacturers. In terms of efficiency, the proposed algorithm can meet the requirements of industrial real-time 3D reconstruction. As can be seen from Figure 20, the proposed method can effectively reduce the GPU memory footprint of voxel hashing [20]. Although the memory footprint of the CPU has increased, it is within a reasonable range. After all, most computer devices support CPU memory expansion, while GPU cannot expand memory capacity. Therefore, our algorithm has high practicability.

Although the proposed algorithm improves the accuracy of thin-surface reconstruction and can meet the scanning requirements of most thin industrial products, the post-processing method of thin-surface topology repair still has some limitations on parameter setting. When the voxel size or truncation width is too large relative to the thickness of the object, the topology-repairing method will lose its effect. It is because our method of topological repair is based on the assumption that some voxels in the distance field are not interfered by the overlap of truncation bands on thin surfaces. The topology repair method uses these unaffected voxels as input to fit the surface and infer the values of the voxels in the affected area. If the voxel size or the truncation width are so large that the truncation bands on opposite sides of the thin surface overlap entirely, then the surface-fitting method loses reliable input, which leads to the inability to reconstruct the accurate surface. Therefore, for thin objects with complex topology, we recommend that users use such parameter configurations as voxel size ≤ 0.5 mm and truncation width ≤ 1 mm. There is no such limitation for thin objects with simple topology, such as a thin plate.

We use hash tables to compress the space and only reconstruct densely near the observed surface to eliminate the memory footprint of the empty space. However, it is not necessary to use regular grids of fixed voxel size to represent all parts of the scene. For example, a desktop needs less detail than a highly structured keyboard on it. Selecting different resolutions to represent different parts of the scene can reduce memory consumption and improve the reconstruction speed at the same time. It is also advantageous to use small voxel sizes to represent areas with complex topology or sharp features. Therefore, in the future, we will study how to build a 3D model with different resolutions adaptively according to the shape of the surface.

## 4. Conclusions

An improved 3D reconstruction method for a thin surface based on a hand-held laser line scanner is proposed. Aiming at the problems existing in the TSDF method for the 3D reconstruction of thin object surfaces, we propose two main improvements: first, in the real-time scanning process, we use the angle between the surface normal and the line of sight to determine the attribution of the thin surface, avoiding the mutual interference between the observation surfaces belonging to different positions; second, we exploit a post-processing method for automatically detecting and repairing topological errors in some areas where misjudgment of thin-surface attribution may occur. Besides, we also improve the underlying data structure of the existing TSDF method to reduce the GPU memory usage. Qualitative and quantitative experimental results provide evidence that the proposed 3D reconstruction approach to a thin surface is valid and can significantly improve the accuracy of thin-surface reconstruction. In comparison with the state-of-the-art laser line scanners, our method shows a similar (C2M distance mean [61]) or even better accuracy (C2M distance Std. [61]). In terms of computing efficiency, the proposed method effectively reduces the GPU memory consumption, which can guarantee a frame rate of more than 60 FPS with the GPU memory footprint under 500 MB.

Although the proposed method can reconstruct a thin surface accurately in most cases, there are still some limitations in parameter setting when reconstructing thin objects with complex topology. If the voxel size or truncation width is too large, mutual interference between the opposite sides of the thin surfaces may still occur. Meanwhile, it is a waste of memory to represent all parts of the scene with a regular grid. In the future, we plan to further improve the robustness and memory efficiency of complex thin-surface reconstruction by adaptively setting different resolutions to the reconstructed model according to the shape of the surface.

## Figures and Tables

**Figure 1 sensors-20-00534-f001:**
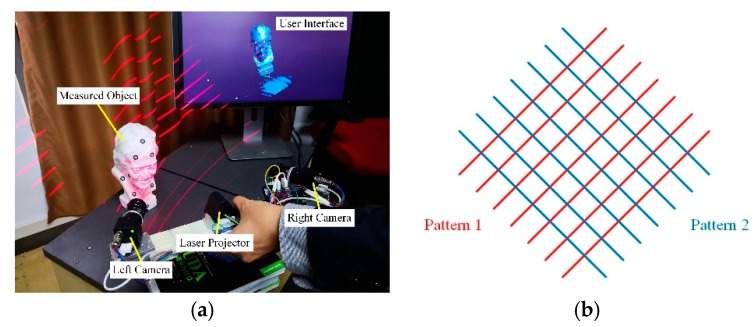
Schematic of the 3D laser scanning system; (**a**) components of the 3D scanner; (**b**) laser patterns; there are two kinds of projection patterns shown as red and blue respectively; we alternately project them onto the measured object; once one of the patterns is extracted, it will switch to the other immediately; the reflection of these two patterns will form multiple intersecting curves on the object.

**Figure 2 sensors-20-00534-f002:**
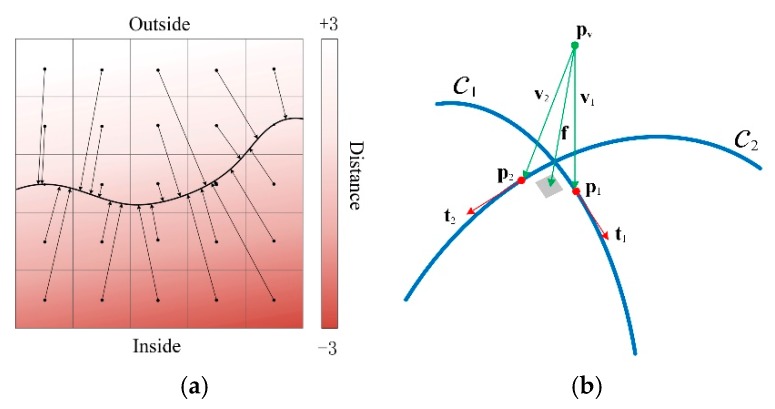
Schematic of truncated signed distance field (TSDF) constructed from intersecting curves; (**a**) a 2D slice of the TSDF computed on a regular voxel grid; each arrow indicates the distance vector of a voxel; gradients of color encode the signs and values of the distances; (**b**) a simple example of computing the value **f** of a vector field.

**Figure 3 sensors-20-00534-f003:**
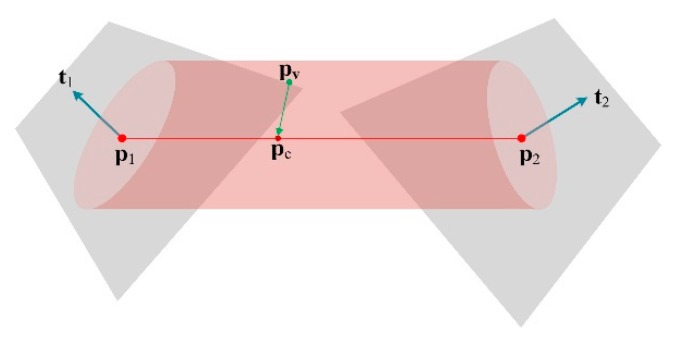
An example of the fundamental cell of the line segment $\bar{p_{1}p_{2}}$; **t**_1_ and **t**_2_ are the tangents at each endpoint of the line segment; all voxels in the cell must be located between the two planes perpendicular to the two tangents and the distance from the central point **p_v_** of voxel **v** toward the closest point **p**_c_ must be less than the cylinder’s radius (envelope size).

**Figure 4 sensors-20-00534-f004:**
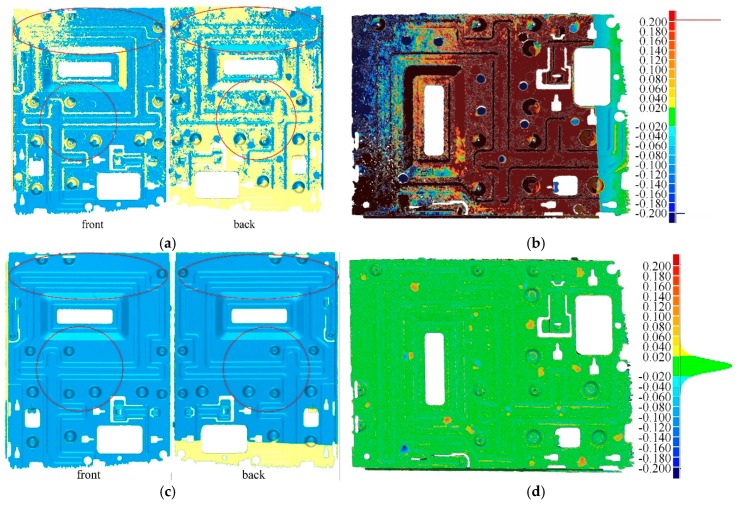
Optimization of the 3D surface reconstruction of a sheet metal part; (**a**) results of Tubić et al. [6]; (**b**) deviation colormap between Tubić’s 3D model and the computer-aided design (CAD) model; (**c**) results of the proposed method; (**d**) deviation colormap of the proposed method.

**Figure 5 sensors-20-00534-f005:**
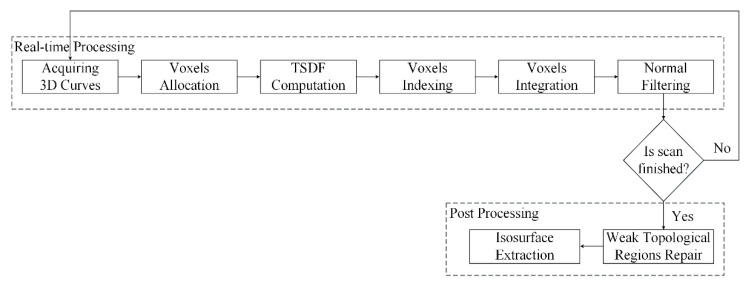
Pipeline of the proposed surface reconstruction method based on 3D curves.

**Figure 6 sensors-20-00534-f006:**
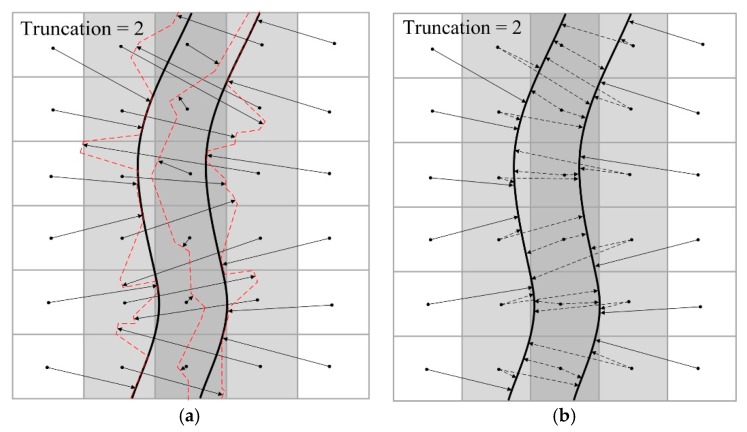
Corrupted thin surface vs. the correct thin surface; (**a**) interfered TSDF and corrupted surfaces; (**b**) correct TSDF.

**Figure 7 sensors-20-00534-f007:**
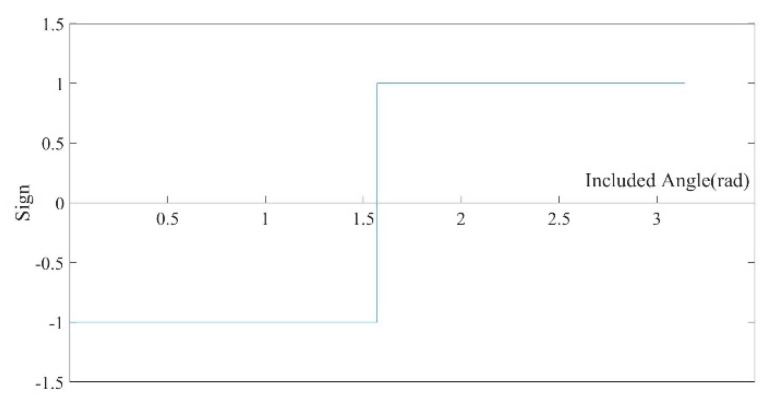
The plot of the sign *S* varying with the angle *α* between vector **n***_dis_* and line of sight **V_v_**; at the angle $=\frac{\pi }{2}$, the sign of TSDF is flipped.

**Figure 8 sensors-20-00534-f008:**
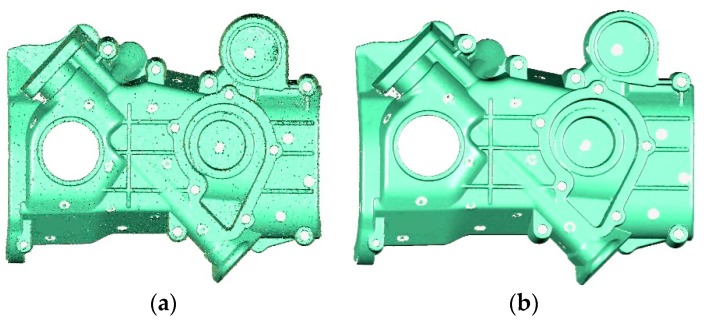
The comparison between the normal computed by Tubić et al. [6] and that filtered by our method; (**a**) normal before filtering; (**b**) normal after filtering.

**Figure 9 sensors-20-00534-f009:**
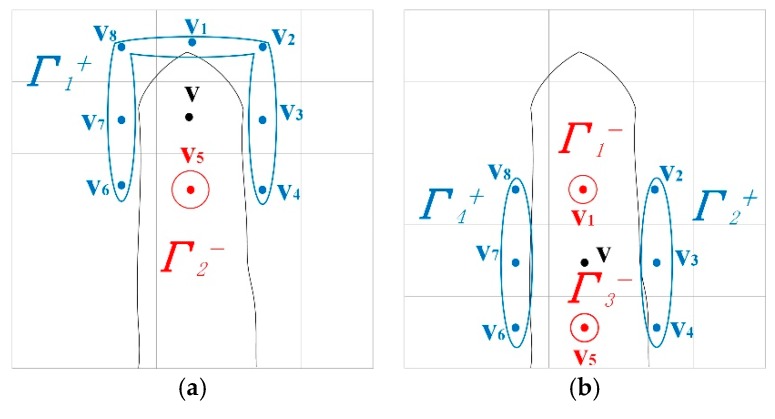
The 2D representation of the critical points identification method; the solid black line represents the actual surface; the neighborhood of voxel **v** is divided into two sets; (**a**) voxel **v**, positive set {**v**_5_}, negative set {**v**_1_, **v**_2_, **v**_3_, **v**_4_, **v**_6_, **v**_7_, **v**_8_}, positive set and negative set constitute two connection groups ***Γ_1_^+^*** and ***Γ_2_^−^*** (red and blue circle), so **v** is not a critical point; (**b**) the voxel **v**, positive set {**v**_2_, **v**_3_, **v**_4_, **v**_6_, **v**_7_, **v**_8_} and negative set {**v**_1_, **v**_5_}; the two sets are divided into four connection groups ***Γ_1_^−^***, ***Γ_2_^+^***, ***Γ_3_^−^***, and ***Γ_4_^+^***, so **v** is a critical point (there are four groups in 2D, but only three in 3D).

**Figure 10 sensors-20-00534-f010:**
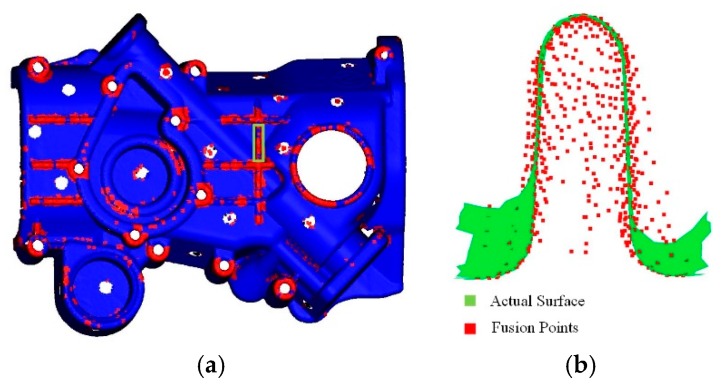
The effect of weak regions detection; (**a**) point clouds are generated from TSDF by Equation (4); the non-critical point is marked blue, and the critical point is marked red; (**b**) a cross-section of a weak region (the green box in Figure (**a**)); there are many points inside the surface that do not fall on the actual surface; these cause the generated surface mesh model to shrink to the inside of the object.

**Figure 11 sensors-20-00534-f011:**
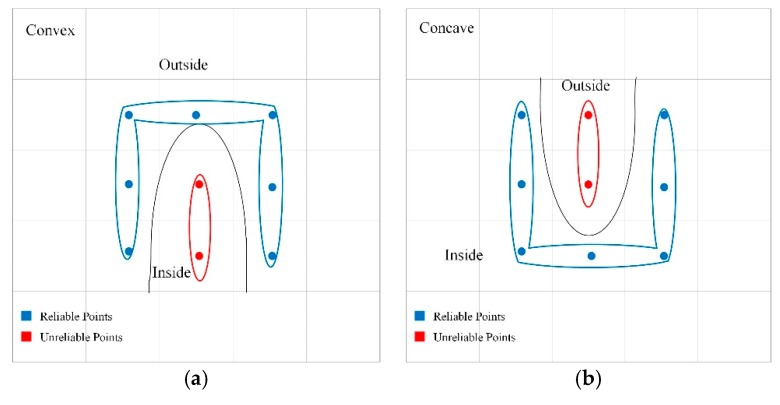
The distribution of reliable and unreliable points in weak regions of convex and concave surfaces; reliable voxel centers are blue, unreliable voxel centers are red; (**a**) distribution of weak regions for convex surfaces; (**b**) distribution of weak regions for concave surfaces.

**Figure 12 sensors-20-00534-f012:**
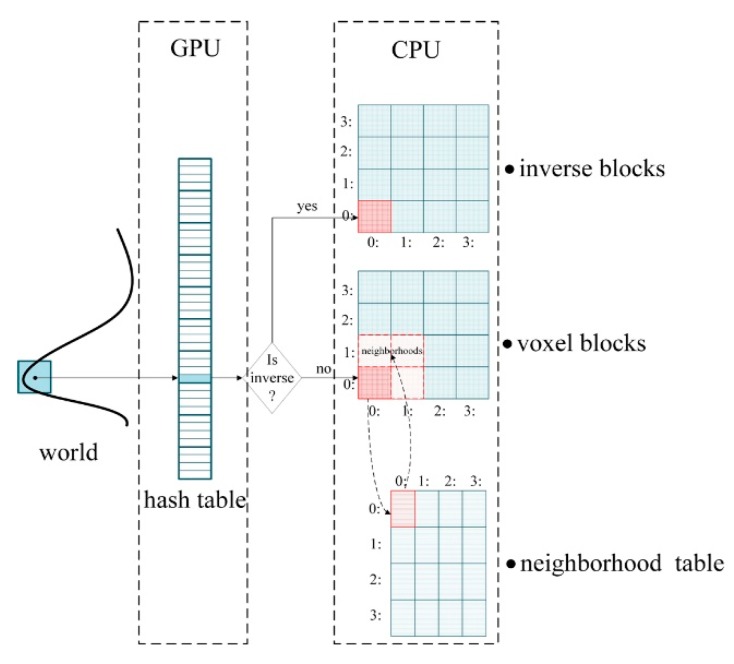
The whole data structure; first, the world coordinates of voxel center are mapped to the hash table to get the address of the voxel block on the central processing unit (CPU), and a thin-surface attribution judgment is then made to store the new TSDFs in either the non-inverse block or the inverse block; the subscripts of the neighborhood lookup table correspond to the individual blocks and each element of the lookup table stores the corresponding neighborhoods of the blocks; only the hash table is stored on the graphics processing unit (GPU), and all other data is stored on the CPU.

**Figure 13 sensors-20-00534-f013:**
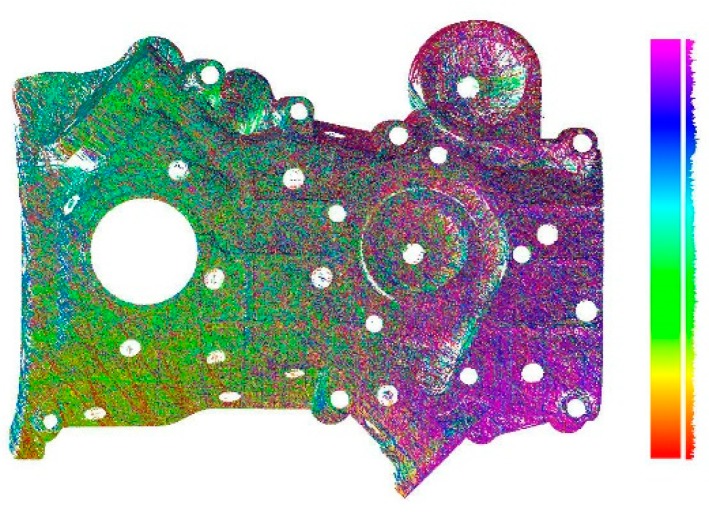
An example of input data of our algorithm; curves are displayed in different colors, and histograms of different color intervals are displayed on the right side.

**Figure 14 sensors-20-00534-f014:**
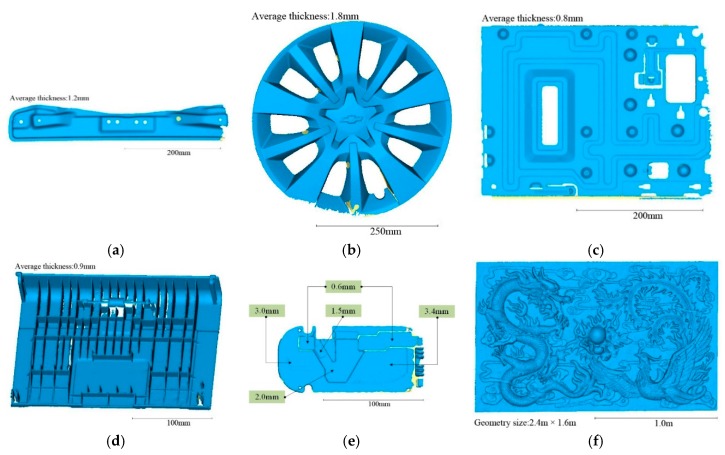
Six experimental objects: (**a**) metal bar; (**b**) automobile hub; (**c**) sheet metal part; (**d**) plastic shell; (**e**) metal panel; and (**f**) relief; the metal panel is composed of five parts with different thickness.

**Figure 15 sensors-20-00534-f015:**
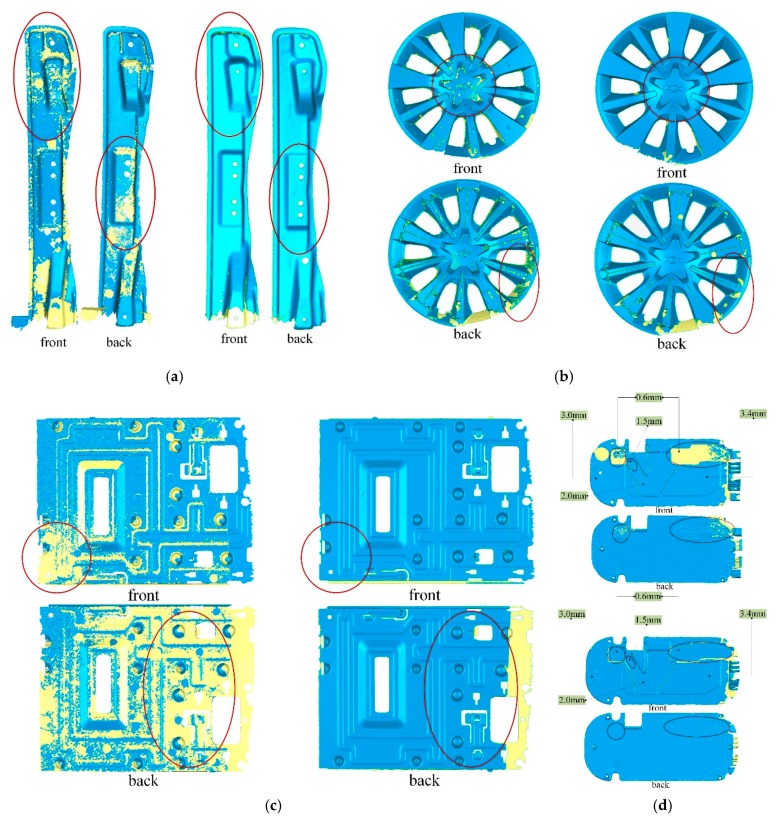
Comparison of scan models from thin products; (**a**) Tubić et al. [6] (**left**) and our (**right**) results of the metal bar; (**b**) Tubić et al. [6] (**left**) and our (**right**) results of the automobile hub; (**c**) Tubić et al. [6] (**left**) and our (**right**) results of the sheet metal part; (**d**) Tubić et al. [6] (**top**) and our (**bottom**) results of the metal panel.

**Figure 16 sensors-20-00534-f016:**
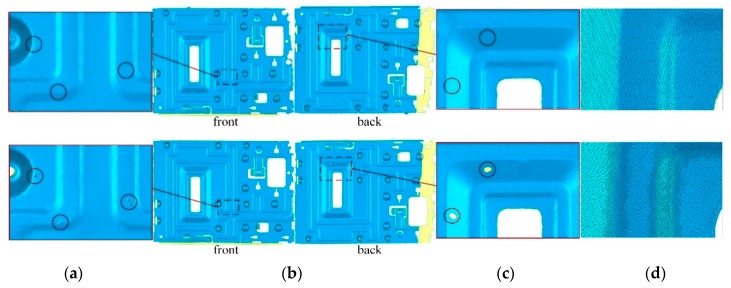
Comparison of our scan results (**top row**) with the Creaform scanner [7] (**bottom row**); (**a**) a comparison of zoomed details at the same location; (**b**) the front and back views; (**c**) the details of another region; (**d**) triangles of the mesh models.

**Figure 17 sensors-20-00534-f017:**
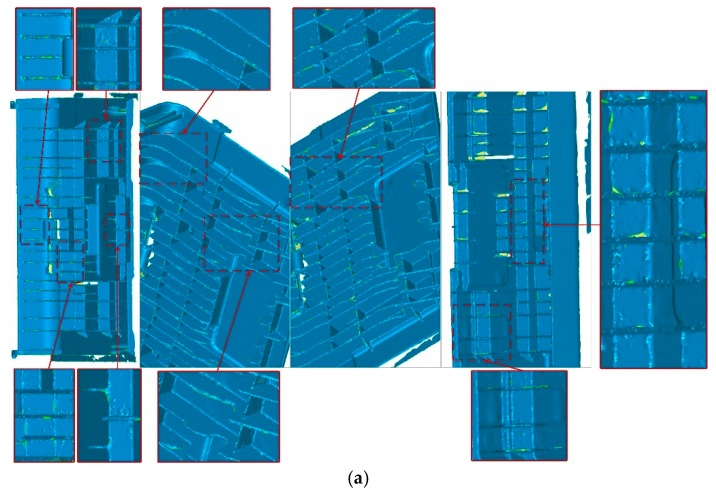
Comparison of the plastic shell model; (**a**) the interference of TSDF brings holes and spikes on the stiffener of the model; (**b**) Creaform’s [7] results show similar problems; (**c**) the proposed post-processing approach repaired the surface of the stiffener.

**Figure 18 sensors-20-00534-f018:**
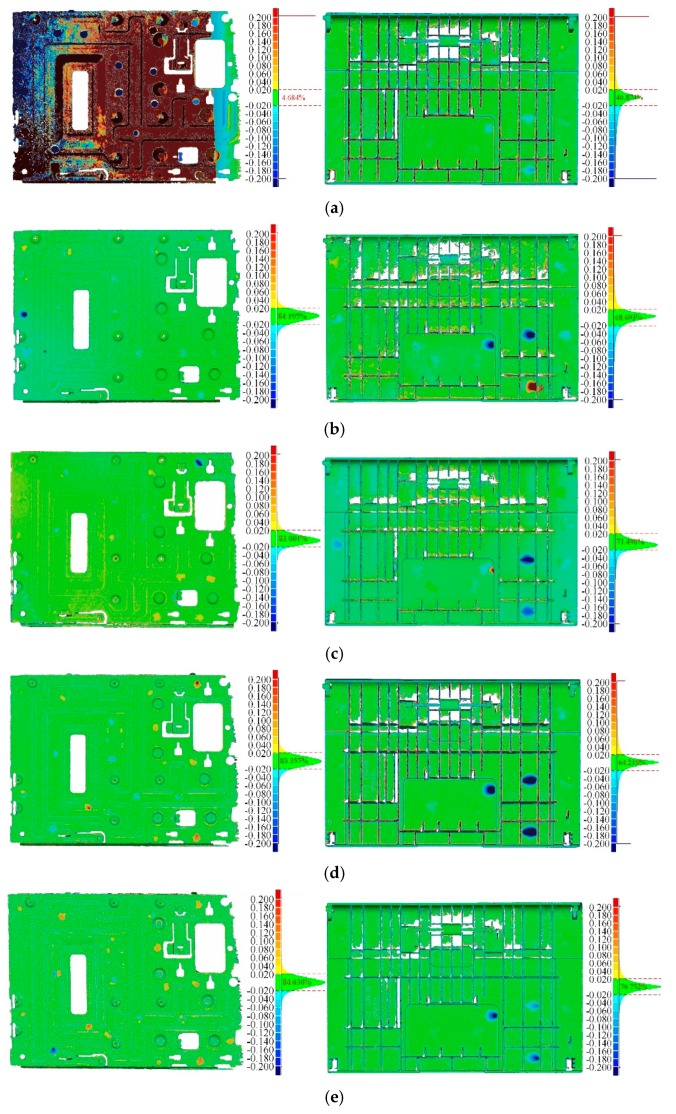
Cloud to mesh (C2M) error colormaps for surface reconstructions of two thin products; the left column is the reconstruction of the sheet metal part, and the right column is the reconstruction of the plastic shell; The compared method are (**a**) Tubić et al. [6]; (**b**) Creaform [7]; (**c**) Scantech [8]; (**d**) our real-time processing approach; and (**e**) our post-processing approach. In each subfigure, frequencies with C2M errors in the interval of [−0.02 mm, 0.02 mm] are also indicated.

**Figure 19 sensors-20-00534-f019:**
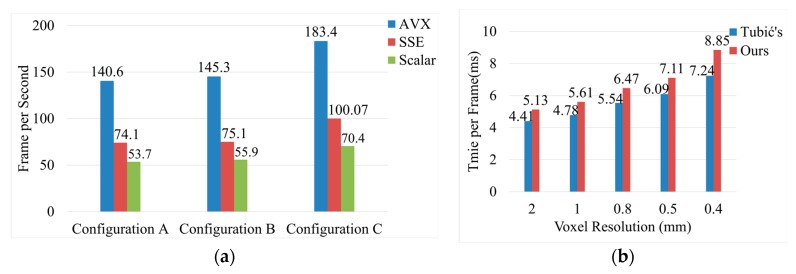
Time efficiency comparison; (**a**) frame rate comparison of real-time reconstruction algorithms on three different hardware configurations in Table 2; the blue bars represent the algorithm implemented with Advanced Vector Extension Instruction Set (AVX) [68], red bars represent Streaming SIMD Extensions Set (SSE) [69], and the green bars represent the version with no instruction sets; (**b**) performance comparison with Tubić et al. [6].

**Figure 20 sensors-20-00534-f020:**
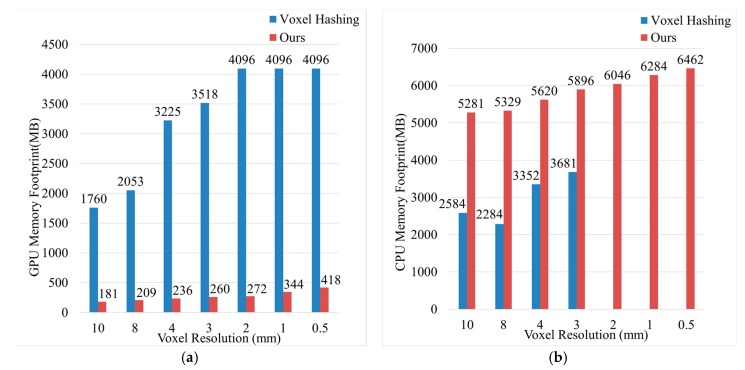
Memory footprint comparisons; (**a**) GPU memory footprint comparisons; (**b**) CPU memory footprint comparisons.

**Table 1 sensors-20-00534-t001:** Accuracy of the scanners involved in the comparison.

Scanner	Volumetric Accuracy ^1^ (Based on Part Size)
Creaform HandySCAN700 [7]	0.020 mm + 0.100 mm/m
Scantech HSCAN771 [8]	0.020 mm + 0.060 mm/m

^1^ ISO 17025 accredited. Visit the manufacturer’s website for more details [7,8].

**Table 2 sensors-20-00534-t002:** Personal computer (PC) configuration for performance test.

PC	Configuration
Configuration A	Intel Core i7-7700HQ (2.8 GHz), 32 GB RAM, NVIDIA GeForce GTX1050 (4 GB)
Configuration B	Intel Core i7-7700 (3.6 GHz), 32 GB RAM, NVIDIA GeForce GTX960 (2 GB)
Configuration C	Intel Core i7-7820X (3.6 GHz), 128 GB RAM, NVIDIA GeForce GTX1060 (4 GB)

**Table 3 sensors-20-00534-t003:** C2M error statistics (mean ± Std.) of different methods in millimeters.

Object	Tubić et al. [6]	Creaform [7]	Scantech [8]	Our Real-Time Approach	Our Post-Processing Approach
sheet metal part	0.1559 ± 0.2592 (544,565) ^1^	−0.0010 ± 0.0750 (1,135,616)	0.0024 ± 0.0408 (620,037)	−0.0011 ± 0.0227 (680,628)	−0.0005 ± 0.0212 (681,574)
plastic shell	0.0093 ± 0.2412 (443,417)	0.0015 ± 0.0913 (441,490)	−0.0053 ± 0.0554 (439,817)	−0.0045 ± 0.0768 (458,489)	−0.0008 ± 0.0379 (489,797)

^1^ The vertices number of the model.

**Table 4 sensors-20-00534-t004:** C2M error statistics (mean ± Std.) for the random samples of different methods.

Object	Tubić et al. [6]	Creaform [7]	Scantech [8]	Our Real-Time Approach	Our Post-Processing Approach
sheet metal part	0.1589 ± 0.2660	−0.0010 ± 0.0737	0.0031 ± 0.0439	−0.0012 ± 0.0231	−0.0011 ± 0.0220
plastic shell	0.0071 ± 0.2727	0.0039 ± 0.1080	−0.0041 ± 0.0691	−0.0032 ± 0.0922	−0.0012 ± 0.0431

**Table 5 sensors-20-00534-t005:** Pearson’s $\chi^{2}$ test for samples of different methods.

Object	Tubić et al. [6]	Creaform [7]	Scantech [8]	Our Real-Time Approach	Our Post-Processing Approach
sheet metal part	0.8562 (3.2968, 7) ^1^	0.3485 (7.8224, 7)	0.5507 (5.9062, 7)	0.7404 (5.1586, 8)	0.6724 (5.7747, 8)
plastic shell	0.9396 (2.9140, 8)	0.5922 (4.6288, 6)	0.9076 (3.3884, 8)	0.4392 (6.9007, 7)	0.4480 (7.8516,8)

^1^ Meaning of values in each cell: *p*-value ($\chi^{2}$ value, degrees of freedom ν).

**Table 6 sensors-20-00534-t006:** Welch’s *t*-test for the sheet metal part.

	Tubić et al. [6]	Creaform [7]	Scantech [8]	Our Real-Time Approach
Creaform [7]	0.0 * (18.0343, 1151) ^1^			
Scantech [8]	0.0 * (18.2162, 1053)	0.7816 (−7.7813, 1628)		
Our Real-time Approach	0.0 * (18.6183, 1014)	0.5343 (−0.0862, 1194)	0.1131 (1.2105, 1514)	
Our Post-processing Approach	0.0 * (18.6405, 1012)	0.5131 (−0.0330, 1175)	0.0956 (1.3072, 1471)	0.4487 (0.1288, 1992)

^1^ Meaning of values in each cell: *p*-value (*t* value, degrees of freedom ν); * significant difference (*p* < 0.05).

**Table 7 sensors-20-00534-t007:** Welch’s *t*-test for the plastic shell.

	Tubić et al. [6]	Creaform [7]	Scantech [8]	Our Real-Time Approach
Creaform [7]	0.0 * (7.2831, 1305) ^1^			
Scantech [8]	0.0 * (7.5697, 1126)	0.5210 (−0.5275, 1699)		
Our Real-time Approach	0.0 * (7.4946, 1224)	0.4410 (0.1483, 1949)	0.4045 (0.2416, 1852)	
Our Post-processing Approach	0.0 * (8.0481, 1049)	0.2307 (7.3644, 1309)	0.1284 (1.1343, 1675)	0.2628 (0.63468, 1416)

^1^ Meaning of values in each cell: *p*-value (*t* value, degrees of freedom ν); * significant difference (*p* < 0.05).

**Table 8 sensors-20-00534-t008:** *F*-test for the sheet metal part.

	Tubić et al. [6]	Creaform [7]	Scantech [8]	Our Real-Time Approach
Creaform [7]	0.0 * (13.0255) ^1^			
Scantech [8]	0.0 * (36.6872)	0.0 * (2.8165)		
Our Real-time Approach	0.0 * (132.0038)	0.0 * (10.1342)	0.0 * (3.5980)	
Our Post-processing Approach	0.0 * (146.0340)	0.0 * (11.2113)	0.0 * (3.9805)	0.0553 (1.1062)

^1^ Meaning of values in each cell: *p*-value (*F* value); the degree of freedom of each term is 999; * significant difference (*p* < 0.05).

**Table 9 sensors-20-00534-t009:** *F*-test for the plastic shell.

	Tubić et al. [6]	Creaform [7]	Scantech [8]	Our Real-Time Approach
Creaform [7]	0.0 * (6.3688) ^1^			
Scantech [8]	0.0 * (15.5554)	0.0 * (2.4424)		
Our Real-time Approach	0.0 * (8.7480)	2.8 × 10^−7^ * (1.3735)	1.0 (0.56237)	
Our Post-processing Approach	0.0 * (39.9106)	0.0 * (6.2665)	0.0 * (2.5656)	0.0 * (4.5622)

^1^ Meaning of values in each cell: *p*-value (*F* value); the degree of freedom of each term is 999; * significant difference (*p* < 0.05).

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
