# Peer review of "Real-Time 3D Reconstruction of Thin Surface Based on Laser Line Scanner"

_sensors, 2020, doi:10.3390/s20020534_

Round 1

Reviewer 1 Report

Manuscript number:  sensors-663605

Title:   Real-time 3D Reconstruction of Thin Surface Based on Laser Line Scanner

Authors:   Yuan He, Shunyi Zheng, Fengbo Zhu, Xia Huang

The Reconstruction of thin surface based on Laser Line Scanners an interesting task. I agree with the concepts explained in the paper.  The mathematical fundamentals and the results are well described. My comments are: the broken regions of a surface have been repaired by mean of laser line projection and parametric functions [1]. Also, the weighted parametric function has been employed to construct a 3D surface [2]. What is the additional improvement?    

[1]   Opt. Eng. Vol. 39 No. 9, p. 2463-2471 (2000).

[2]   Opt. Eng. Vol. 53, p. 092003, (2014).  

Reviewer 2 Report

The paper intends to focus on TSDF development.  The authors propose three improvements to the existing TSDF methods.  The paper is interesting and the reviewer would suggest the following items to be addressed before any further step:

Line 23: how effective? be specific in clarifying the contribution of the work.

Line 24: Support the claim of 0.02 mm in the discussion section and cite other works with more than 0.02 mm accuracy. Compare their work and context. For example in lines: 656, 659, 661, 669. This section starts, from 655 needs to be compared with the literature. 

Section 1 needs to be improved by citing new papers. The voxel literature is limited but the following papers can be useful to be cited since they are fresh papers:

"Visualising the urban green volume: Exploring LiDAR voxels with tangible technologies and virtual models." Landscape and urban planning 178 (2018): 248-260.

Spatial compactness metrics and Constrained Voxel Automata development for analyzing 3D densification and applying to point clouds: A synthetic review. Automation in Construction96, pp.236-249: 

https://www.sciencedirect.com/science/article/pii/S0926580518305326

Pixels, voxels, and views: A study of shape representations for single view 3D object shape prediction. In Proceedings of the IEEE Conference on Computer Vision and Pattern Recognition (pp. 3061-3069):

http://openaccess.thecvf.com/content_cvpr_2018/html/Shin_Pixels_Voxels_and_CVPR_2018_paper.html

Improve the conclusion section by addressing the followings: contributions of the study, technical limitations, future work in specific tasks. 

Reviewer 3 Report

In this manuscript, a method for reconstruction of a thin surface based on a hand-held laser scanner is presented. It is an improvement of the Truncated Signed Distance Fields (TSDF) computation method. It consists of seven steps shown in fig. 5 and explained in the text. The method is evaluated qualitatively and quantitatively, in my opinion, in a convincing way. I do not find any major issues. As for the presentation layer, I think that the captions of some figures are too long. In most cases, the same information is repeated in the main text. Therefore, if possible, I would suggest shortening these captions. I would also add a summary with information on how many different objects were used for the method validation.

Round 2

Reviewer 2 Report

There are many statements inserted into the paper which is good.  However, the correctness of results (all the newly added statements) and the contribution of the paper claimed in Sections 3.1 onwards, can be a serious issue from the readers perspective.  In order to conclude if the proposed improvement is acceptable, the comparison of both scanners mentioned in lines 556 and 557 should be based on some criteria/metrics. All details should be comparably shown in a table, e.g. the information mentioned in lines 556 to 575, etc.

In addition, t-test/ANOVA or similar comparison tests should be applied and reported to claim the differences. Otherwise, the significance of the differences can not be reported. Table 1 or Figure 19 is useful but the results here can not be judged. 

The discussion section is the explanation of results and tables. However, discussion means that the author should discuss how different are the results with the current literature, previous practices, etc. There is no reference mentioned in Section 3.5. This will reduce the scientific aspect of the paper. 

Author Response

This manuscript is a resubmission of an earlier submission. The following is a list of the peer review reports and author responses from that submission.

Round 1

Reviewer 1 Report

This paper presents a 3D point cloud post-processing method for the hand-hold multi-line laser 3D scanner. The article is generally well written and organized. However, the author should clearly identify which 3D laser scanner they are experimented. For the post-processing, like normal smooth and noise removal etc., should have been a common function and tool in the supporting software provided with the scanner. Only comparison with a method in [9, 2004] is not convinced obviously. In some figures like Fig. 14, normal in some areas are inverse. Does that affect the processing? In addition, some grammar errors should be double-checked. Generally, novelty of the methodology is quite limited and results are not convinced. Some CG related journals are suggested for this submission.

Reviewer 2 Report

Summary:

The paper presents methods to improve Truncated Signed Distance Fields (TSDF) methods by solving the problem of interference between opposite surfaces using thin surface attribution judgement. Moreover, the authors propose a framework that integrates GPU and CPU resources so as to ensure the real-time performance. Qualitative results on different types of thin objects have been provided along with the quantitative results to show the speed and memory footprint.

Comments:

+ The paper presents a thorough and detailed account of the proposed approach.

+ The qualitative results shown in figures 14-19 show impressive performance of the proposed method.

- The paper seems like an engineering approach to address a special challenge with TSDF instead of a hypothesis derived research. It is a combination of different steps which are taken to solve the problems which were encountered during the reconstruction.

- It is not clear how 'T' is selected in section 2.1

- In figure 3, it is not evident how p1 and p2 are the closest points to voxel v on curves c1 and c2.

- What is the vector between v1 and v2 in figure 3.

- The caption of figure 3 is repeated from the description in section 2.1

- Explanation for equation 1 should be provided.

- The first paragraph of section 2.2 is repeated.

- The reproducibility of the proposed approach is in question. The paper comprises quite many heuristics and minor details which makes it extremely difficult to implement. I would suggest authors to reorganize the paper as well as to provide a list of all the steps in the proposed method.

- It would be better to explain lines 226-232, using an image/illustration if possible.

- The manuscript should be revised to correct grammatical errors.

- It is difficult to know the novelty in section 2.3.1 and it looks like an explanation of an existing approach. I would recommend summarizing this section and providing relevant references for further study.

- I would suggest revising section 2.4 and 2.5 to remove redundancy and only include relevant details.

- For figures 14, 15, 17 and 18, it would be useful to show the improvements from the proposed approach by marks and arrows.

- In line 532, it should be ‘Figure 20’ instead of ‘Figure 18’.

- The paper lacks any quantitative evaluation for the 3D reconstruction. It is difficult to make any definitive statement about the improvement without any quantitative evaluation.

- It is very difficult to read any text and numbers in figures 20-22.

- I would suggest adding a discussion section which can outline drawbacks, solutions and future work in detail.

Round 2

Reviewer 1 Report

To my experience, the post processing described by the paper and the registration of thin parts, have been well solved by some off-the-shelf 3D hand-hold laser scanner, like the Creaform Handyscan etc. The authors should clearly prove and demonstrate the improvement of their system or algorithms, at least with a comparison to these usual instruments. 

Reviewer 2 Report

Although the manuscript has been improved, however, it still remains a difficult read and challenging to follow. I would recommend a few other suggestions:

The caption for figure 3 should be revised as it is a verbatim of the text in the preceding section. There are still quite a few unexplained (redundant) equations. Equations 18-21 should be explained.